# Design and Analysis Method of Piezoelectric Liquid Driving Device with Elastic External Displacement

**DOI:** 10.3390/mi15040523

**Published:** 2024-04-13

**Authors:** Wangxin Li, Mingfeng Ge, Ruihao Jia, Xin Zhao, Hailiang Zhao, Chuanhe Dong

**Affiliations:** 1Jinan Guoke Medical Technology Development Co., Ltd., Jinan 250001, China; jiarh@sibet.ac.cn (R.J.); zhaoxin_dyx@163.com (X.Z.); zhaohailiang@sibet.ac.cn (H.Z.); dongchuanhe@sibet.ac.cn (C.D.); 2Suzhou Institute of Biomedical Engineering and Technology, Chinese Academy of Sciences, Suzhou 215163, China; gemf@sibet.ac.cn; 3School of Biomedical Engineering, University of Science and Technology of China, Hefei 230000, China

**Keywords:** piezoelectric, elastic force, numerical calculation, fluid-structure interaction

## Abstract

In piezoelectric drive, resonant drive is an important driving mode in which the external elastic force and electric drive signal are the key factors. In this paper, the effects of the coupling of external elastic force and liquid parameters with the structure on the vibrator resonance frequency and liquid drive are analyzed by numerical simulation. The fluid-structure coupling model for numerical analysis of the elastic force was established, the principle of microdroplet generation and the coupling method of the elastic force were studied, and the changes in the resonant frequency and mode induced by the changes in the liquid parameters in different cavities were analyzed. Through the coupled simulation and calculation of the pressure and deformation of the cavity, the laser vibration measurement test was carried out to test the effect of the vibration mode analysis. The driving model of the fluid jet driven by the elastic force on the piezoelectric drive was further established. The changing shape of the fluid jet under different elastic forces was analyzed, and the influence law of the external elastic force on the change in the droplet separation was determined. It provides reference support for further external microcontrol of droplet motion.

## 1. Introduction

Based on the piezoelectric response micro-displacement technology, the micro-electro-mechanical fusion principle is used to achieve precision on-demand controllable microfluidic manipulation and control technology. Piezoelectric microfluidic generation technology is widely used. The earliest piezoelectric inkjet technology was widely researched and developed in the 1980s and developed piezoelectric inkjet equipment to supply piezoelectric equipment to Epson [1]. With the continuous progress of MEMS technology [2,3,4], microdroplet control technology has been widely used in precision machining technology [5,6] and bio-MEMS technology [7,8]. Piezoelectric response is highly controllable due to its high precision. This technology has been widely studied and applied in piezoelectric brakes and ultrasonic motors, and the motion resolution of non-resonant excitation drivers based on piezoelectric drivers can reach nm level [9], with the advantages of large stroke and high load. German Bulent Delibas [10] and others have developed single-source and double-source double-frequency resonance (DSDFR) ultrasonic excitation. The piezoelectric motor has reached a maximum no-load speed of more than 220 mm/s and a push-pull capacity of 2.5 N, while the piezoelectric linear ultrasonic motor has been rapidly developed with a compact structure [11]. It has been widely used in precision liquid control instruments [12,13], microelectronics [14], biomedicine [15,16], and single-cell micromanipulation [17,18], featuring high resolution and sensitivity, greatly improving the controllable degrees of freedom, and realizing a low error integration of complex equipment and processes.

The microfluidic vibration generated by the piezoelectric effect can generate the driving force of the liquid, which is the core of the microfluidic motion control operation. The driving process and motion characteristics of the fluid can be effectively controlled by itself combined with the fluid characteristics and vibration mode coupling. At present, the injection effect can be effectively controlled based on liquid materials and control signal parameters, and parameters such as the velocity and volume of droplets can be controlled within a certain range [19,20,21,22].

At present, the piezoelectric material is used as the controller for the generation control of microdroplets, which is mainly controlled by the control signal; the generation volume and motion parameters of the droplet are controlled. Pyungho Shin et al. [23] studied the excitation control of piezoelectricity nozzle of extrusion type, and tested the control of low viscosity liquid by single pulse and double pulse excitation methods with voltage and pulse interval time as the main control parameters. Sofija Vulgarakis Minov et al. [24] studied the mode of on-demand droplet (DOD) generation driven by piezoelectricity, pulse width and pulse amplitude in DOD mode, and the effect of frequency and pulse amplitude in continuous mode on droplet characteristics. The droplet velocity in DOD mode ranges from 0.08 to 1.8 m/s. Ti-Yuan Shan et al. [25] proposed a piezoelectric microjet device, and studied the influence of the voltage and pulse frequency of the driving signal on the injection effect. Ning Liu et al. [26] proposed a trapezoidal waveform design method for extruding piezoelectric inkjet printhead to provide improved stable injection and optimal droplet shape. San Kim [27] proposed a method to analyze the influence of ink supply pressure on the injection effect, which is a method to explore the rule of injection effect except for the control signal. Tian Jiao et al. [28] analyzed how voltage and signal frequency controlled droplet formation speed and size. Roberto Bernasconi et al. [29] realized the generation of high-viscosity droplets through piezoelectric drive signals. Yuming Feng et al. [30] proposed a method for controlling metal droplets through piezoelectric excitation waveform, and verified the duty cycle and amplitude methods. Young-Jae Kim et al. [31] controlled droplet generation by electric field intensity and signal frequency. However, in the control process of piezoelectric injection technology, the liquid injection speed and volume are mainly sprayed through excitation signals, material parameters, etc., and there is a lack of research on the external load force of the piezoelectric vibrator to control the injection. On the basis of excitation signal control, it is necessary to combine the external elastic force to increase the control range of injection volume and speed, and the control method of droplet forming is necessary.

On the basis of numerical simulation and laser vibration measurement experiment, a design and analysis method of a piezoelectric liquid driving device with elastic force is presented in this paper. Firstly, the axial vibration driving model is established based on the spring elastic force and piezoelectric vibrator. Secondly, the resonance analysis of the elastic force and the influence of the cavity material parameters are numerically simulated to determine the resonance influence law and the influence effect of the material parameters; the laser vibration measurement test is carried out to analyze the excitation vibration mode. Finally, the external elastic force is analyzed and simulated on the droplet-forming motion parameters. The combination of different wave elastic forces and excitation signals can produce different droplet-forming effects, which can enhance the control of droplet jet forming and form a variety of droplet-forming control methods.

## 2. Model and Principle Structure Design

The single layer PZT-5 material used in Figure 1 can serve as the main body of the piezoelectric actuator, which can be driven by a single layer of the piezoelectric actuator. It mainly includes a nozzle, a fixed seat, a spring actuator, a ceramic fixing ring, and an insulation plug. The relevant parameters of nozzle structure size are shown in Table 1. The liquid driving principle of piezoelectric driver is shown in Figure 1a. The structure diagram is shown in Figure 1b.

The principle of piezoelectric vibrator drive is mainly that the vibrator produces different resonant responses under the action of the excitation signal. The sealing structure ensures that the pressure of the cavity is not affected by air fluctuations. The material of the fixed seat and the nozzle is cemented carbide aluminum, and the insulating plug is the insulating material PVC. The design model diagram of the piezoelectric fluid driving principle diagram and overall elastic damping microfluidic structure diagram is shown in Figure 1. Under the active excitation effect of its own elastic damping, this structure is used as the source of fluid driving force in the cavity, and the mechanical modal characteristics of the structure itself realize fluid driving according to the harmonic response effect.
(1)$$S=s_{E}T+d^{T}E$$

The elastic stiffness matrix is $S_{E}$, $S_{E}=\left[\begin{array}{cccccc} 16.4 & -5.74 & -7.22 & 0 & 0 & 0\\ -5.74 & 16.4 & -7.22 & 0 & 0 & 0\\ -7.22 & -7.22 & 18.8 & 0 & 0 & 0\\ 0 & 0 & 0 & 47.5 & 0 & 0\\ 0 & 0 & 0 & 0 & 47.5 & 2\\ 0 & 0 & 0 & 0 & 0 & 44.3 \end{array}\right]\times 10^{-12}\;m^{2}/N$.

Piezoelectric coupling coefficient, $d_{T}=\left[\begin{array}{cccccc} 0 & 0 & 0 & 0 & 584 & 0\\ 0 & 0 & 0 & 584 & 0 & 0\\ -171 & -171 & 374 & 0 & 0 & 0 \end{array}\right]\times 10^{-12}\;C/N$.

Relative dielectric constant, $\frac{\varepsilon_{T}}{\varepsilon_{0}}=\left[\begin{array}{ccc} 1730 & 0 & 0\\ 0 & 1730 & 0\\ 0 & 0 & 1730 \end{array}\right]$.

Vacuum dielectric constant, $\varepsilon_{0}=8.854\times 10^{-12}$ F/m.

## 3. Numerical Simulation and Analysis

### 3.1. Principle of Numerical Analysis

Under the action of the excitation signal, the piezoelectric oscillator generates vibrations and under the combined action of elastic force, different resonant states are generated between its own vibration and the natural frequency of the structure, then different vibration effects are generated. Different vibration material parameters and different geometric structures produce different vibration effects. In this paper, the modal response under the action of elastic and inelastic external forces is studied; the structural modal response of different cavity materials is studied, and a numerical simulation method is proposed. The structure and operation principle of the whole device are shown in Figure 2. The main generation of piezoelectric droplets works in the resonant state of the whole device. Its main working principle process is that on the basis of elastic force, the electrical signal is mainly used to generate piezoelectric vibration, combined with the inherent modal frequency of the structure and then the resonant state of the structure is generated, and different liquid driving modes are generated. Different vibration modes have different driving effects on the liquid. At the same time, in the resonant state, the driving force of the elastic force is increased to enhance the driving force amplitude. Then the volume and velocity range of the droplet control can be expanded. As shown in Figure 2a, the initial state is as shown in Figure 2a. With no elastic pressure and electric signal drive, the back pressure of the cavity is balanced with the liquid viscosity and capillary resistance reaching a stable state. No driving force is generated, and the cavity is mainly waiting for the injection of liquid. At the same time, the elastic force *F* is increased on the outer surface of the piezoelectric oscillator, as shown in Figure 2. The size of *F* depends on $\Delta x_{0}$ the beginning. The *k* is the elastic coefficient of the spring, which can be considered as a constant value in the process of nano-scale deformation of the piezoelectric oscillator. The elastic force generated is shown in the Equation (Equation 2).
(2)$$F_{0}=-k\cdot \Delta x_{0}$$

Under the action of the initial elastic force, the initial deformation of the piezoelectric oscillator is shown in Figure 2b. The bending deformation $\Delta x_{f}$ is shown by the Equation (Equation 3). The liquid is initially extruded under the action of the elastic force $F_{0}$. The surface tension of the liquid increases, where EI(x) is the flexural stiffness. The flexural stiffness varies with the radial dimension of the circular oscillator. M(x) is the bending moment generated by the interface under the action of $F_{0}$, which changes with the radial dimension.
(3)$$EI\left(x\right)\Delta x_{f}=\int_{l}\left(\int_{l}M\left(x\right)dx\right)dx+C_{1}x+C_{0}$$

When the liquid state exhibits a specific promoting tension, as depicted in Figure 2c, a driving signal is applied to the piezoelectric oscillator. This driving signal can be generated using various waveforms such as sine wave, trapezoid wave, and rectangular wave, thereby achieving diverse driving effects. As shown in Equation (Equation 4), under the action of a driving signal with period *T*. The time of $t_{1}$ is the signal amplitude, and the oscillator is driven along the polarization direction of the piezoelectric oscillator. The piezoelectric oscillator then generates a deformation of $\Delta x_{s}$, such as the Equation (Equation 4). At this time, the total shape variable of the piezoelectric oscillator is Δx; meanwhile, the liquid is driven and the liquid drop has a certain velocity.
(4)$$\Delta x=\Delta x_{f}+\Delta x_{s}$$

As shown in Figure 2d, at the moment $t_{2}$, the electric field of the oscillator is 0 or the electric field direction of the oscillator is opposite to the polarization direction. This will cause the center point of the oscillator to move in the opposite direction of the elastic force. In addition, the deceleration is contrary to the initial velocity, and the vibrator displacement generates $\Delta x_{d}$ in Figure 2b; compared with the previous time $t_{1}$, the vibrator at this time returns to the original state changing from the maximum deformation to the minimum. At this time, the volume of the cavity increases, resulting in the pressure in the cavity becoming smaller, which causes the fluid to split and form a liquid drop. Thus, reducing the pressure in the cavity and the pressure at the outlet and realizing the movement tearing of the droplet at the outlet; realizing the generation of the droplet, and completing the entire liquid motion state under the action of elastic damping motion. Under the action of micro-displacement, the elastic coefficient of the spring is unchanged, $F_{0}$ is affected by the amount of deformation, and the $F_{0}$ of the initial state is constant, and its initial state is determined by $\Delta x_{0}$.

The elastic coefficient of the spring is 1.2 N/cm, the density of the copper gasket is $8.9\times 10^{3}\;\mathrm{kg}/m^{3}$, and the external shell structure is made of lightweight aluminum with a density of $2.7\times 10^{3}\;\mathrm{kg}/m^{3}$. In the process of vibration driving, the PZT-5 piezoelectric plate, copper plate and fixed structure all have a certain influence on the modal analysis of the structure. The modal response effect plays a key role in the control of pressure waves, the selection of excitation signals, and the setting of motion mechanical characteristics. The coupling analysis model was established under the effect of the fluid-structure coupling effect and the combined simulation effect of elastic damping. The overall structure size is shown in Figure 3. Based on the three-dimensional coupling model, the harmonic response was calculated and analyzed. The mode and response were numerically calculated and simulated.

### 3.2. Numerical Analysis Calculation

The interaction between fluid and solid is studied in the Lagrangian coordinate system by means of the fluid-structure coupling finite element model. In this paper, the fluid-structure coupling model is used to analyze the dynamic characteristics of the fluid under the action of elastic force. Analysis in terms of linear elastomer and fluid viscosity and compressible ideal liquid. Global system structure dynamics equations like (Equations (Equation 5) and (Equation 6)):(5)$$M_{s}\ddot{\mu }+C_{s}\dot{\mu }+K_{s}\mu =F_{s}$$
(6)$$\left(M_{s}+M_{a}\right)\ddot{\mu }+K_{s}\mu =0$$

The mass matrix of the system in the formula; $M_{s}$ is the mass matrix of the system, $M_{a}$ is the fluid mass additional matrix, $\ddot{\mu }$ is the acceleration vector of the system node, $C_{s}$ is the damping matrix of the system, $\dot{\mu }$ is the velocity vector of the system node, $K_{s}$ is the stiffness matrix of the system, and μ is the displacement vector of the node. $F_{s}$ is the external elastic damping. Where $F_{s}$ is composed of the elastic force generated by the external spring, which is composed of Equation (Equation 1) $F_{0}$. In the basis of the structural dynamics equation, the fluid-structure coupling equation is introduced, and the fluid control Navier–Stokes equation is as follows, Equations (Equation 7) and (Equation 8):(7)$$\frac{\partial \rho_{t}}{\partial t}+\nabla \left(\rho_{t}v\right)=0$$
(8)$$\frac{\partial \rho_{t}}{\partial t}+\nabla \left(\rho_{t}v-\tau_{t}\right)=f_{t}$$

The type of $\rho_{t}$ as the fluid density, *t* for time, *v* as the velocity vector, $\tau_{t}$ for the shear stress tensor, and $f_{t}$ for fluid volume force. Due to the physical characteristics of different liquids, they have different effects on the modal effects of the jet structure. The calculation and analysis of the effect of the fluid-structure coupling simulation are mainly affected by the mechanical properties of the liquid structure, sound velocity, density and viscosity. In this paper, according to the different material characteristics of liquid water and air, the parameters are set and the modal and harmonic response of the three-dimensional coupling model are calculated and analyzed. The numerical calculation model is established to calculate and analyze the structural modal.

The results of structural modal analysis in air state and fluid state are shown in Figure 4 and Figure 5. As shown in the figure, there is no obvious change in the 1–2 order vibration modes of the structure, but there is a great change in the third order vibration modes, and the vibration modes are different. The resonant frequency of the cavity itself has a relative offset, and the oscillator has different effects when there are different media in the cavity. The resonance frequency data of the wet mode compared with the dry mode are shown in Table 2. When the liquid in the cavity is water, the first-order mode frequency is 5.086 kHz due to the influence of liquid parameters. In the case of the same mode, the resonance frequency of the wet mode decreases to a certain extent under the action of the viscosity and density of the liquid. The first-order resonance frequency of the structural modes in the air state in the cavity is 10.632 kHz, and the first-order mode frequency increases. The wet mode has a certain influence on the free vibration of the structure itself. The wet mode frequency in the low mode is 50% in the dry mode, and the basic wet mode is 37% in the high mode. It can be seen that the cavity fluid has a great influence on the modal effect of the structure when the mode is low. It has the effect of reducing the natural frequency of the structural mode. Therefore, the frequency of the wet mode effect needs to be reduced when the resonance response is applied.

Under the influence of different materials in the cavity, the harmonic response of the cavity structure is analyzed under the condition of a vacuum, air and common liquid water. Different parameters of the materials in the cavity have different offset effects on the resonant frequency of the overall structure and have different effects on the resonant amplitude. The resonance effect of the structure is affected by different cavity materials. Combined with the modal effect in the vacuum state, the external pressure load effect is the spring static pressure effect, which acts on the axis perpendicular to the contact surface. The force influences the preloading effect of the piezoelectric vibrator. Modal changes occur under external forces. The 1–3 mode vibration effect in the air state is shown in Figure 4a–c; the 1–3 mode analysis of the fluid-structure coupling mode effect in the liquid state is shown in Figure 5a–c. Under the self-damping effect of the liquid, the harmonic response of the fluid is generated under the excitation of the harmonic signal. Due to the influence of the viscosity and density parameters of the liquid, the harmonic response effect is different. As shown in Figure 6a, when the cavity is full of water and air, it can be seen that the harmonic response frequency of the cavity varies under the action of different materials. The results show that under the action of water, the first-order resonant frequency of the cavity is the lowest, and the first-order resonant frequency of the structure in the air state is 10.6 kHz. It can be seen that different materials in the cavity have obvious effects on the resonant state of the structure, and the resonant coupling effects of different fluid materials are obtained based on the fluid-structure coupling analysis. Figure 6b shows the influence of the oscillator amplitudes in the resonant and non-resonant states of air and liquid water, K is the slope of the fitting line for the amplitude, and it is evident that the fitting slope for the resonant amplitude is higher than that for the nonresonant amplitude, indicating that the cavity pressure increases significantly with amplitude under resonant conditions. At the same time, a 5 kHz excitation signal is applied to obtain the first-order resonant frequency in the liquid state and the transient changes in liquid pressure in the cavity under the action of the non-resonant state, as shown in Figure 6c,d. The pressure changes in the air cavity are chaotic. The effect of the liquid cavity is stable and obvious at the first-order resonant frequency.

Under the effect of external elastic force, the vibration response of the piezoelectric vibrator is directional to the vibrator and the external elastic force. Due to the compression of the spring, the piezoelectric oscillator vibrates in the same direction as the spring, increasing the amplitude. When the vibrator vibrates outside the cavity, the spring elastic force is opposite to the direction of vibration. Under the influence of the elastic force, the harmonic response analysis results are shown in Figure 7a,b. Under the influence of the elastic force, the overall resonant frequency of the external elastic force does not change greatly, the outlet pressure increases in the fixed natural frequency, and the radial deformation size also changes greatly in amplitude. The outlet pressure and the external elastic force basically increased linearly (as shown in Figure 7c). The linearity remained high.

Piezoelectric oscillators produce forced vibrations under the action of elastic forces. The nonlinear vibration of the oscillator under the action of the electrical signal and mechanical force signal. The forced vibration of the electric signal and force signal directly acts on the piezoelectric oscillator. By establishing the damping model of external load, the corresponding unsteady force and axial force are set up. In the resonant state, the axial force (as shown in Figure 8a) and the piezoelectric oscillator driving signal (as shown in Figure 8b) are numerically stimulated to simulate the whole structure. The vibration changes in the oscillator are analyzed. As shown in Figure 8c, the pressure at the outlet of the cavity changes under the action of the first-order mode frequency driving signal and the action of external changing force, it is obvious that the outlet pressure and external pressure basically increase linearly, the linear fitting variance reaches 0.99, and the linear slope is 0.20776. As shown in Figure 9a, it can be concluded that under the action of the peak value of the external variable force. The peak value of the oscillator amplitude gradually increases with the action of the external variable force up to 50 µm. As shown in Figure 9b, the change in the outlet pressure under the action of the external variable force, the superposition of the force signal and the electrical signal can increase the peak value of the outlet pressure under the action of the same frequency variable load.

After analyzing the influence state of structural resonance of different liquid materials, it is determined that different materials do have a certain influence on the resonance state. Because the influence of different materials on the mode has multiple factors, the density and viscosity factors on the resonance effect are analyzed, and the vibration influence analysis under the condition of a single density is carried out. In the density range of 600–1200 $\mathrm{kg}/m^{3}$, with the increase in density and the stability of other parameters, the analysis results show that the resonant frequency decreases with the increase in density. As shown in Figure 10a of the axial vibration analysis and the results of radial analysis, it can be seen that density change significantly changes the resonant frequency of the oscillator. Therefore, when setting the calculations and the analysis for the materials with the same density, different resonant frequency controllable vibration modes are adopted.

## 4. Experimental Device and Method

In order to verify the accuracy of numerical simulation, the results of the coupling effect of the microfluidic piezoelectric fluid structure were demonstrated and tested. We conducted testing and simulation of fluid vibration materials and tested the effects of different materials such as air and water in the cavity. We tested the resonance state and set up an injection instrument to measure the harmonic response mode of the structure through laser vibration measurement. The liquid is supplied to the cavity through the liquid tube to keep the liquid filled in the cavity with a certain back pressure. The back pressure can be adjusted by an external force setting. At the same time, the back pressure has a force effect on the piezoelectric oscillator and produces a certain force response together with the elastic force. The driver signal of the oscillator is composed of a signal generator, a computer and a power amplifier. The undistorted excitation signal is amplified through the power amplifier to realize the driver of the excitation signal required by the oscillator. To monitor the response of the vibrator in the structure, a high-frequency laser vibrometer and oscilloscope are used to monitor the excitation signal. Combined with computer data processing software, the vibration mode of the structure under different materials and different frequencies and other parameters of the excitation signal are synchronized with the detection of the laser vibrometer and a unified connection is adopted to achieve the same time detection. The principle structure diagram of the overall test is shown in Figure 11.

The experimental platform mainly consists of an oscilloscope (Tektronix MS02024B), signal generator (AFG31152), power amplifier (PA0) and laser vibrometer (Polytec). The experimental results of laser vibration measurement can provide a reference for the numerical simulation results and guarantee the correctness of the numerical analysis model. In the process of measurement, the side of the micro-jet structure is fixed by a fixed fixture, the piezoelectric oscillator is driven by a resonant wave, and the mode of the oscillator is measured. In this experiment, different liquid effects of water and air were selected for vibration measurement. According to the liquid vibration measurement object, the modal response of the structure is calculated and analyzed, and the correctness of the calculation method is verified. According to the setting of the boundary conditions of the inlet and outlet of the cavity, the amplitude of the voltage is gradually increased and the frequency is gradually increased to achieve the ultrasonic state. Under the structural analysis of each mode, the structural amplitude under the resonant state is tested and analyzed, and the resonance effect is calculated and analyzed.

In this study, the laser vibration measurement test of the whole experimental device was carried out to measure the resonance effect of the structure, and the resonant frequencies of each order of the fluid drive device were calculated to provide a certain reference for the resonance drive. At the same time, the vibration measurement effect of the structure was calculated and analyzed under the characteristic parameters of different materials. The signal generator provides the driver signal with an adjustable frequency which is driven by different signal waveforms, e.g., sine wave, rectangular wave and trapezoid wave. The power amplifier amplifies the signal by 15 times. Under the premise of not losing the fact, the oscillator and the laser vitiometer measure the vibration synchronously, record the vibration mode of the oscillator, and collect the response data of the oscillator. The resonance effect and modal response of the piezoelectric oscillator were recorded. The setting of the laser vibration analysis experiment is based on the whole.

## 5. Results and Discussion

The overall test device is shown in Figure 12. The experimental results on the influence of structural fluid effects on different materials based on different parameters and characteristics. Alloy steel is the main material, and laser vibration measurement experiments are conducted on the structure in the fluid region. Water and air are present in the cavity and different elastic loads are applied for testing. The vibration effect and cavity mode of the structure are analyzed. The laser vibration measurement effect of the vibration is shown in the figure, and the core parameter of the resonant mode is the resonant frequency. In the test of natural frequency, the modes of each order are shown in Figure 13a without the action of load, and in Figure 13b with the action of load; it can be seen that the effect of the experimental results is the same as that of the simulation results. The deviation of the resonant frequency of the first mode is 1%.

The vibration measurement effect after modal analysis is shown in Figure 14. The overall vibration mode is shown in Figure 14a–d. The test shows that under the first-order mode vibration analysis state, it can be concluded that in the case of laser vibration measurement, the entire object mainly vibrates axially with a large central amplitude. The overall vibration mode is single, and the cavity is almost squeezed parallel. Squeezing generates a driving force that drives the movement of the liquid.

The results of the second-order mode modes are shown in Figure 15. The modes are not parallel at the same time, but there are simultaneous vibrations of two different structures. When the structure vibrates, the pressure in the cavity is not uniformly distributed as a whole; when the structure vibrates and the vibration modes are not parallel, the pressure wave transmission in the cavity is unstable, and the droplet drive will produce unstable motion. The liquid is mainly concentrated in the periodic movement of the cavity, and the axial movement of the pressure concentration cannot be realized, and the liquid ejection of the nozzle cannot be realized.

In the first-order excitation oscillator test and simulation basis, the excitation signal is determined and the structural microjet is simulated. The results show that the resonant frequency and mode of the oscillator have a certain influence on the fluid driving under the action of an external elastic force. The driving state of the fluid under the state of piezoelectric-solid coupling was calculated and simulated. Under the action of elastic force and inelastic force, the pressure at the nozzle was shown in Figure 16a. Under the action of different elastic forces of 0 N, 5 N and 10 N, the pressure of the nozzle was simulated and analyzed under the application of 0.5 ms periodic elastic force and 100 V step signal. The changing trend of the nozzle pressure under the drive of external elastic force remains unchanged, but under the action of external elastic force, the pressure generated in the same direction changes greatly, and the external elastic force has no influence on the change in the overall pressure trend. The maximum peak value of the nozzle pressure driven by the peak elastic force of 10 N can reach 80 kPa, and the minimum peak value of the nozzle pressure under the case of no elastic force (0 N) is 10 kPa. Due to the inertia resonance of the oscillator, the vibration amplitude of the oscillator will increase under the action of the external elastic force at the same frequency, and then the pressure difference of the liquid injection chamber will increase. The peak value of the outlet pressure is sensitive to the effect of the external elastic force, and the effect on the motion is great. The simulation of the liquid motion state under different nozzle pressure changes shows that the droplet driving force is small under the condition of no elastic force (0 N), and the droplet quantity and velocity are small. The simulation results show that (*t* = 0.175 ms) will produce separation and fracture, and the liquid separation time will increase under the increase in the same frequency elastic force. When elastic force (*F* = 5 N), the separation time is 0.2 ms, and when elastic force (*F* = 10 N), the separation time is basically 0.2 ms. Under the action of the elastic force, the speed increases with the increase in the elastic force, and the maximum speed generated when the elastic force is large is 81 m/s. The analysis results show that with the increase in the external elastic force, the liquid motion speed increases, and the time interval of droplet forming separation increases. The simulation results also show that the tail droplet length increases with the increase in the elastic force, and the single volume of the droplet increases. The peak value of the external elastic force should be controlled under the condition that the volume and velocity of the droplet are guaranteed. The overall simulation results show that under the premise of a fixed piezoelectric signal, the external elastic force drives the droplet ejection forward, and the same driving direction increases the nozzle pressure, thus increasing the droplet ejection speed and the forming volume of a single droplet and the droplet tail stretch increases.

In the study of external elastic pressure inlet pressure and injection molding, the external rectangular wave elastic force is taken as the effective elastic excitation load and the frequency excitation is 5 kHz. The droplet shape generated by the 5 kHz excitation waveform is analyzed. The regularity of the elastic force waveform is analyzed on the basis of the fixed excitation waveform condition. The analysis results are shown in Figure 17. The same frequency first-order resonance frequency is 5.01 kHz and the peak elastic force is 5 N to produce the droplet forming effect. On the basis of duty cycle 1:1, the droplet forming state corresponding to the rectangular wave (Figure 17a), triangular wave (Figure 17b) and sine wave (Figure 17c) is shown. It can be clearly seen that in the control of the waveform, the difference in droplet separation state appears. The rectangular wave vibrating droplet (Figure 17d) has a large volume and high speed, and the droplet is separated at 0.145 ms. Compared with the triangular wave and sine wave, the droplet separation time is short and the movement speed is high, which itself is related to the high average value of the rectangular wave pressure. The separation time of a triangular wave (Figure 17e) is 0.152 ms, which has a longer separation time and a smaller droplet volume. When driven by the external superposition of the sine wave (Figure 17f), the droplets are separated at 0.150 ms, and the volume size and movement speed of droplet are in the middle.

As shown in Figure 18a, the velocity variation of a droplet under a single cycle is shown in Figure 18b. The peak velocity is related to the external elastic force, and the maximum value 10 N corresponds to the maximum velocity peak, followed by the peak value 5 N which corresponds to the elastic force. As shown in Figure 18b, the volume of a droplet generated by a single complete cycle has a certain positive correlation with the external elastic force. The larger the volume of a single droplet is, the droplet volume is at the microliter level. The increase in the external elastic force also causes a change in droplet volume and running speed, which also provides a way to enhance the controllability of the piezoelectric drive.

## 6. Conclusions

Based on the research of piezoelectric-current-solid coupling, the external elastic force based on the piezoelectric excitation signal is simulated and analyzed. A piezoelectric fluid-solid coupling structure under the action of elastic force is proposed. The piezoelectric-fluid-elastic coupling analysis is carried out to explore its action law. A numerical simulation method is proposed to study the multi-order vibration modes of piezoelectric vibration under the action of external elastic force and the pressure formation law under the multi-parameter effects of different materials and densities in the cavity. The vibration mode verification is carried out by laser vibration measurement test. The numerical simulation shows that the external elastic force has no obvious effect on the vibrator mode and resonant frequency, and the resonance effect is mainly controlled by the internal material parameters. The external elastic force causes the exit pressure to increase linearly. Numerical simulation was conducted to study the formation, splitting, tail length generation and movement of droplets with the increase in the amplitude of external elastic force below 10 N. Under the action of the same frequency elastic force, the movement velocity of a droplet and the volume of a single droplet showed a linear increase with the increase in the amplitude. When the splitting time was above 5 N, the splitting time point basically coincided. The droplet-forming states under the action of different waveforms are also significantly different, and the single droplet splitting forming time is more consistent. This research analysis mainly proposed a law of the effect of external elastic force on the basis of traditional electric excitation signals, and the research proved that the external elastic force and the excitation signal have a great influence on the formation and movement speed of the droplet under the same frequency. However, in the actual realization of the external controllable active multi-type elastic force method, further exploration is needed.

## Figures and Tables

**Figure 1 micromachines-15-00523-f001:**
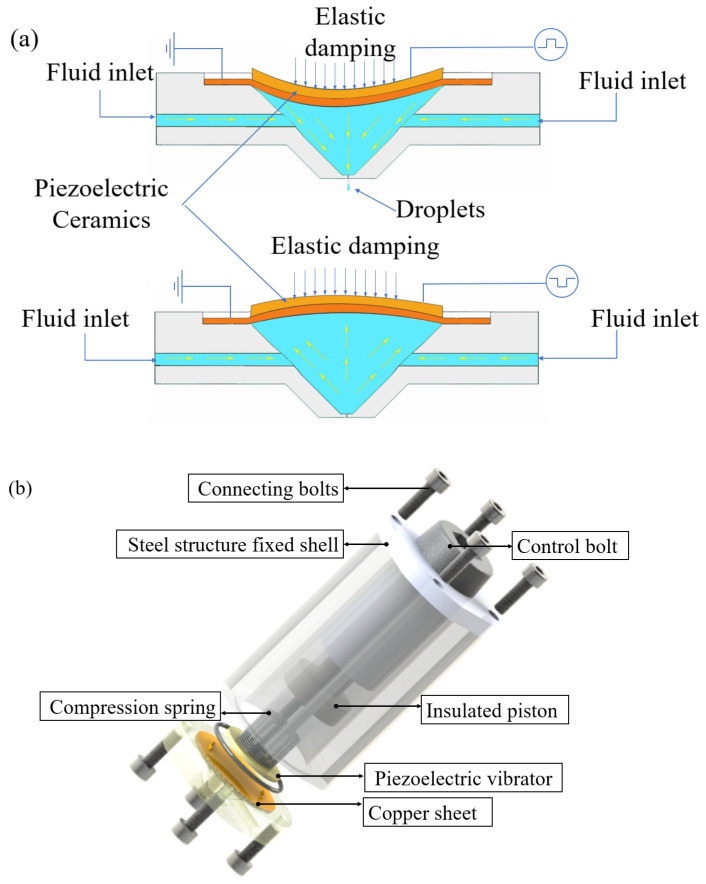
Global structure model. (**a**) Structure schematic diagram. (**b**) Jet structure model.

**Figure 2 micromachines-15-00523-f002:**
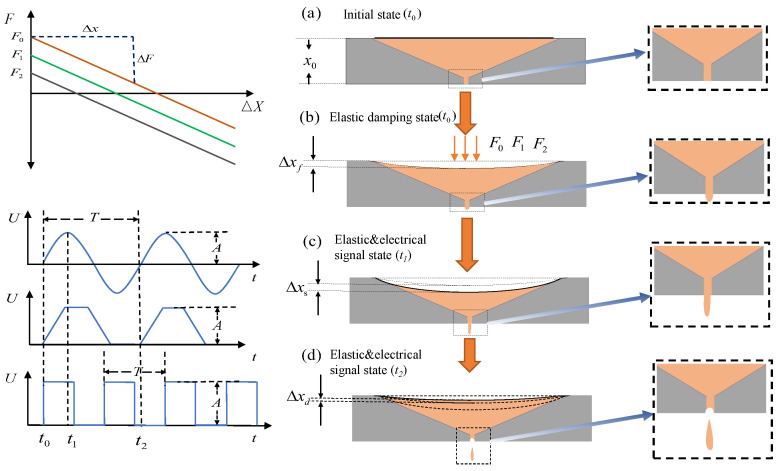
Structure schematic diagram.

**Figure 3 micromachines-15-00523-f003:**
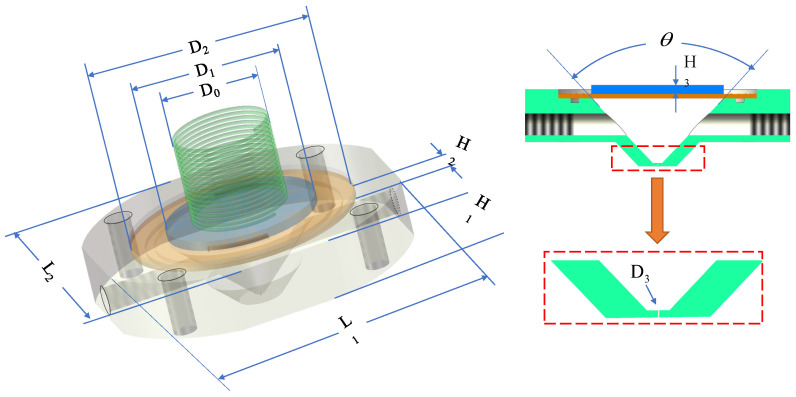
Dimension structure of fluid-structure coupling harmonic response analysis model.

**Figure 4 micromachines-15-00523-f004:**
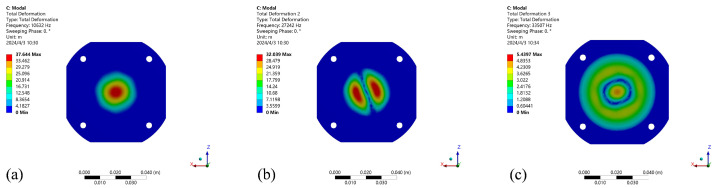
Analysis results of 1–3 modes in cavity air state. (**a**) 1-mode. (**b**) 2-mode. (**c**) 3-mode.

**Figure 5 micromachines-15-00523-f005:**
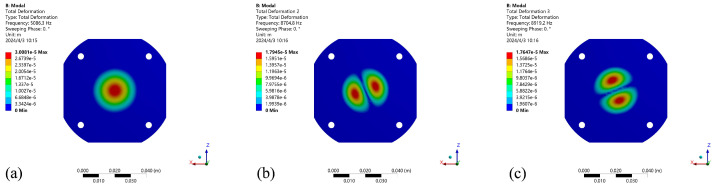
Analysis results of 1–3 order modes. (**a**) 1-order mode. (**b**) 2-order mode. (**c**) 3-order mode in cavity liquid water state.

**Figure 6 micromachines-15-00523-f006:**
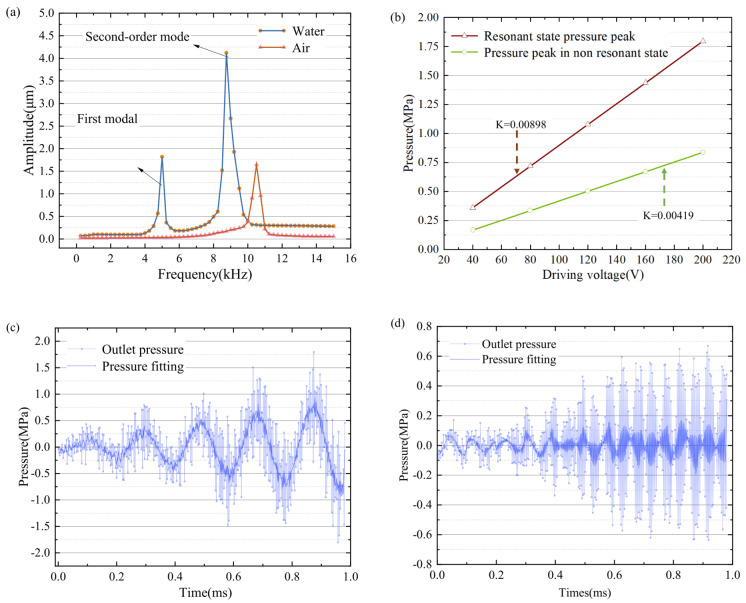
(**a**) Influence of resonant frequencies of liquid water and air. (**b**) Influence of resonant and non-resonant states. (**c**) Transient analysis of resonant outlet pressure. (**d**) Transient analysis of non-resonant outlet pressure.

**Figure 7 micromachines-15-00523-f007:**
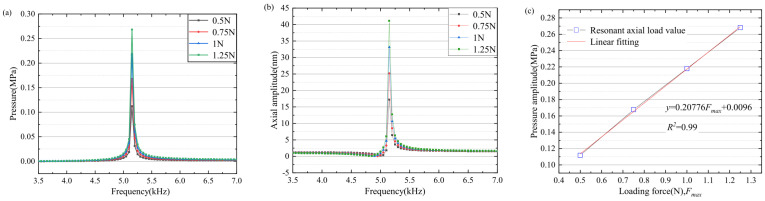
(**a**) Outlet pressure. (**b**) Axial deformation. (**c**) The force is the same as the pressure amplitude of the nozzle.

**Figure 8 micromachines-15-00523-f008:**
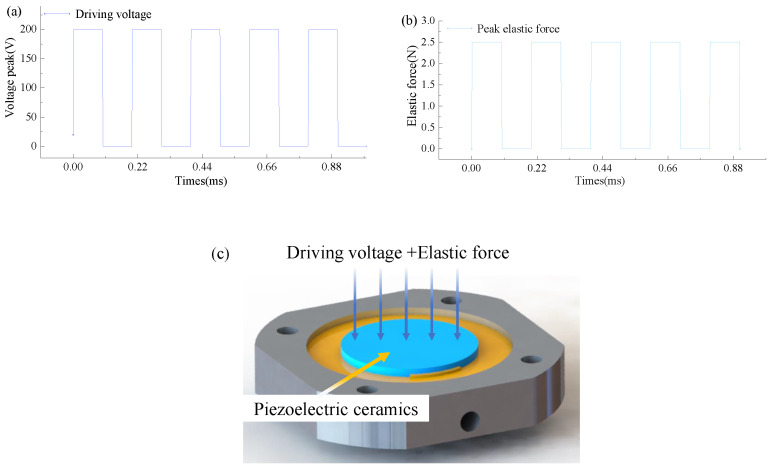
(**a**) Piezoelectric vibrator drive signal. (**b**) External variable axial force. (**c**) Signal application diagram.

**Figure 9 micromachines-15-00523-f009:**
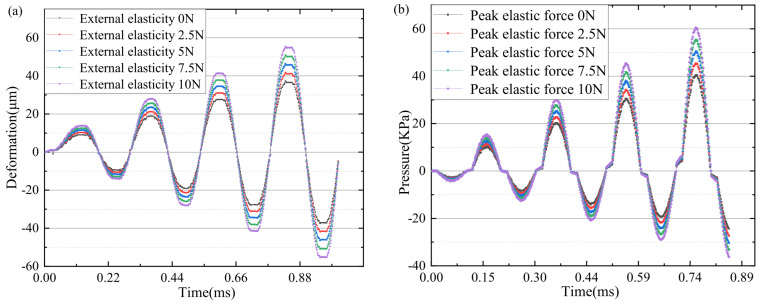
Vibrator deformation and end pressure under different external amplitude. (**a**) Piezoelectric vibrator axial deformation. (**b**) Nozzle outlet end pressure.

**Figure 10 micromachines-15-00523-f010:**
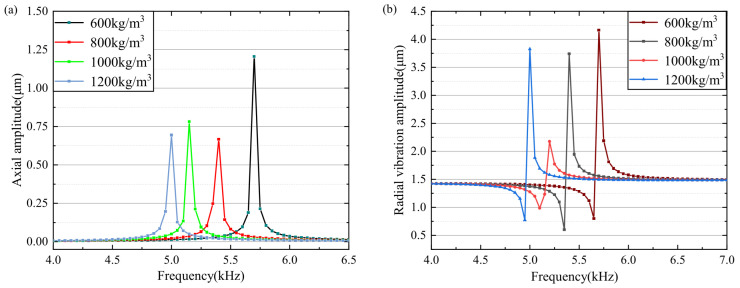
Material density parameter analysis. (**a**) density change axial deformation. (**b**) density change radial deformation.

**Figure 11 micromachines-15-00523-f011:**
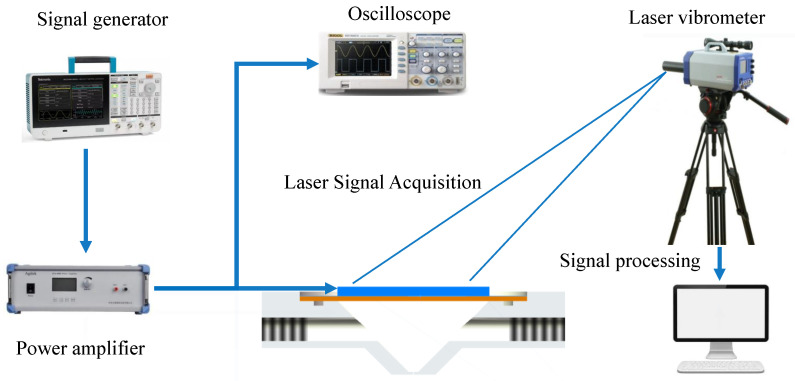
Laser vibration measurement test principle diagram.

**Figure 12 micromachines-15-00523-f012:**
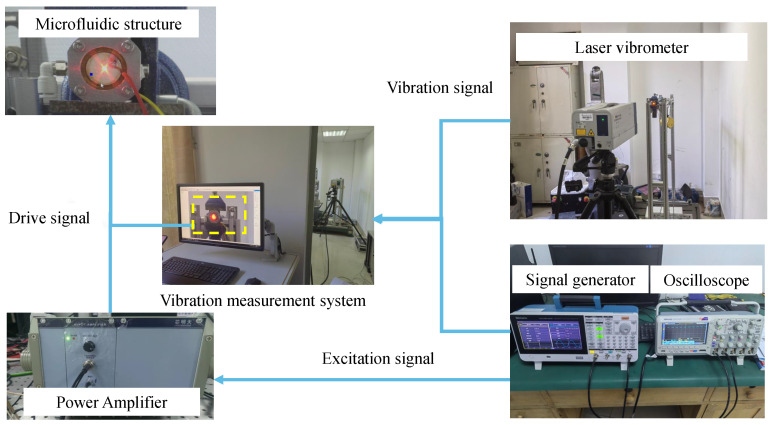
Laser vibration measurement test.

**Figure 13 micromachines-15-00523-f013:**
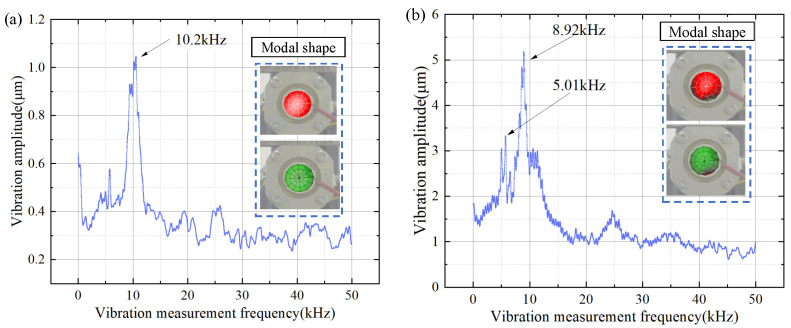
Laser vibration test results. (**a**) Cavity air vibration test results. (**b**) Cavity liquid water laser vibration test results.

**Figure 14 micromachines-15-00523-f014:**
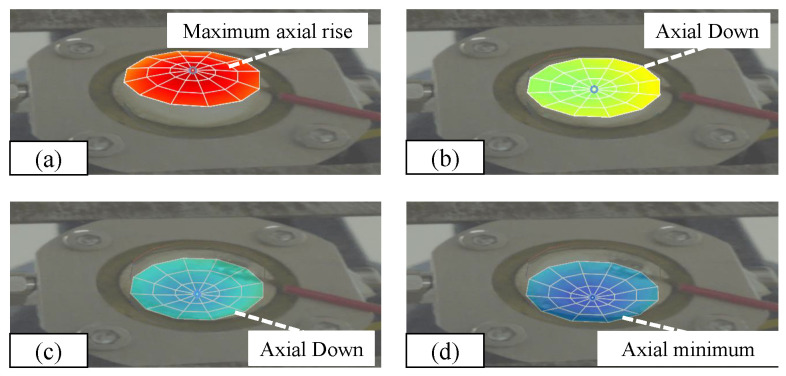
(**a**) Maximum axial amplitude. (**b**) Downward axial vibration. (**c**) Downward axial vibration. (**d**) Lowest axial amplitude.

**Figure 15 micromachines-15-00523-f015:**
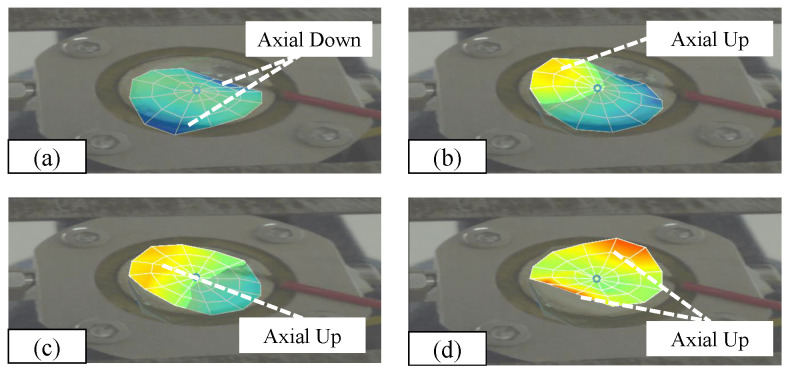
(**a**) Two-side axial downward vibration. (**b**) One-side upward vibration. (**c**) One-side upward vibration. (**d**) Two-side upward vibration.

**Figure 16 micromachines-15-00523-f016:**
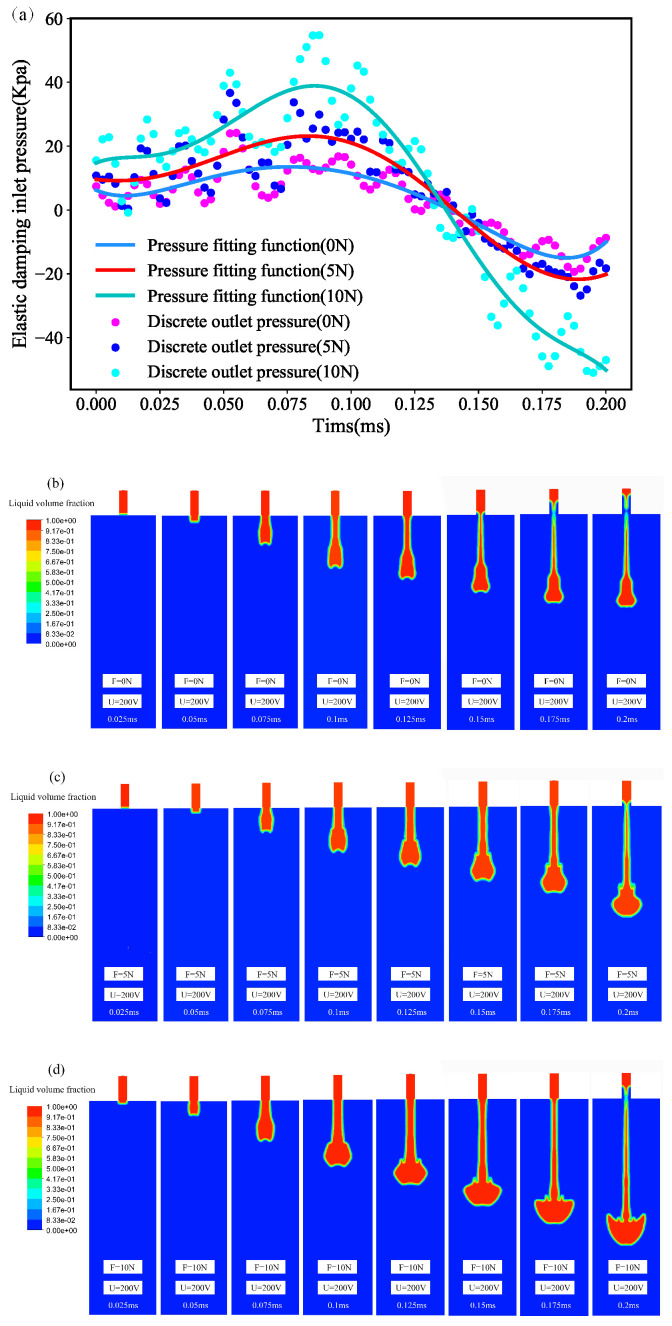
(**a**) Nozzle pressure change. (**b**) Droplet formation under inelastic conditions. (**c**) Droplet formation under 5 N elastic condition. (**d**) Droplet formation under 10 N elastic condition.

**Figure 17 micromachines-15-00523-f017:**
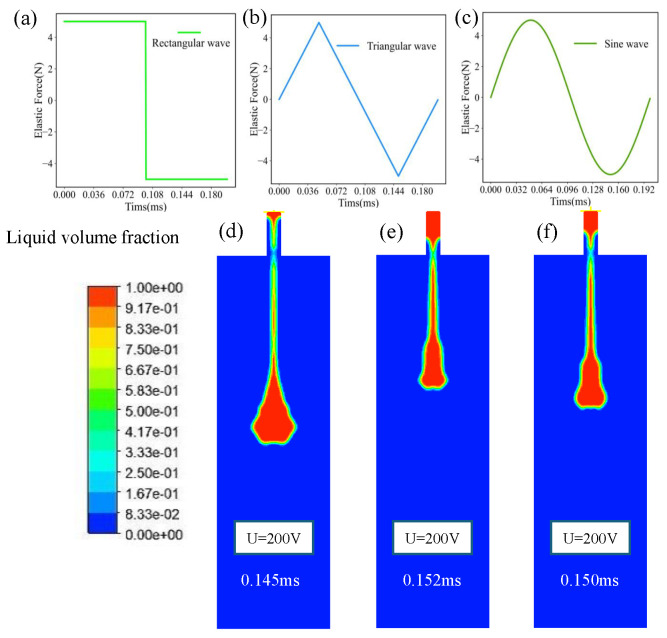
(**a**) Rectangular wave. (**b**) triangular wave. (**c**) Sine wave. (**d**) Rectangular wave liquid motion. (**e**) Triangular wave liquid motion. (**f**) Sine wave liquid drive.

**Figure 18 micromachines-15-00523-f018:**
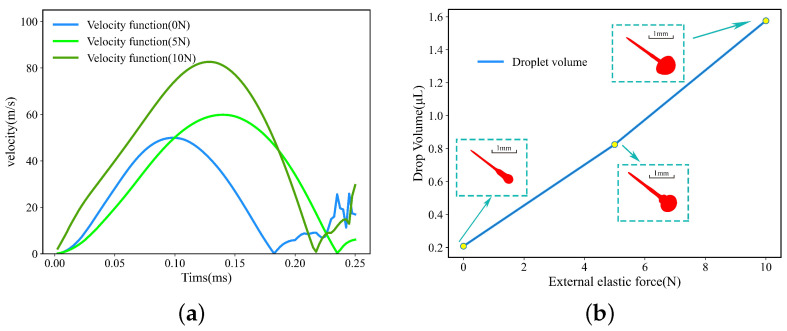
(**a**) Velocity variation. (**b**) Droplet volume change.

**Table 1 micromachines-15-00523-t001:** Elastic force piezoelectric microfluidic drive structure dimension parameter table.

Structural Parameters	Values	Structural Parameters	Values
$D_{0}$	9 mm	$L_{1}$	32 mm
$D_{1}$	16 mm	$L_{2}$	32 mm
$D_{2}$	24 mm	$H_{1}$	6 mm
$D_{3}$	0.1 mm	$H_{2}$	0.5 mm
θ	90°	$H_{3}$	1.5 mm

**Table 2 micromachines-15-00523-t002:** Calculation of resonant frequencies of different cavity materials.

Mode of Vibration	Mode 1	Mode 2	Mode 3	Mode 4	Mode 5	Mode 6
water (kHz)	5.086	7.704	7.919	15.092	15.316	16.224
Air (kHz)	10.632	27.242	33.507	36.631	41.016	43.372

## Data Availability

The original contributions presented in the study are included in the article, further inquiries can be directed to the corresponding author.

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
