# Peer review of "Design and Analysis Method of Piezoelectric Liquid Driving Device with Elastic External Displacement"

_micromachines, 2024, doi:10.3390/mi15040523_

Round 1

Reviewer 1 Report

Comments and Suggestions for Authors

Dear Authors,

The manuscript should be revised before it is published. My remarks are listed in the attachment.

Kind Regards.

Comments on the Quality of English Language

There are plenty of mistakes related to the grammatic. Extensive editing of English language required. 

Author Response

Dear Reviewer

        Thank you very much for taking the time to review this manuscript.Please take a look at my revised draft, see the attachment.

Reviewer 2 Report

Comments and Suggestions for Authors

1. Please polished the paper to make it more readable. 

2. There are some errors in the paper, for example:

(1) Repeated description in lines 72-76

(2) Where is t2 in Figure 2?

(3) Is Δxd described in fig.2d? (in line 129) Please mark precisely and explain how Δxd is formed?

(4) Is F0 in equation 2? (in line 15) 

3. “the results of the piezo-acoustic coupling effect of microfluidics were demonstrated and tested” as description in lines 250.

Is the acoustic providing the vibration frequency in the simulation? If so, please provide a description of the simulation model. 

4. How is the external elastic force applied through the structure? Is it supplied through the compression of the spring by the insulated piston? Please explain the source of the external elastic force before Fig. 1. 

5. Confusing description of experimental conditions.

Fi6(b) shows the influence of oscillator amplitudes in resonant and non-resonant states of air and liquid water.

Is it at 5kHz that the air cavity is non-resonant and the liquid cavity is resonant? if so, Fig. 7 shows the results for the liquid cavity? 

6.The outlet pressure increases with time, as shown in Fig. 9. Does the changing pressure have any effect on the volume and velocity of the ejected droplets?

 7. The deformation is at the fourth peak about 4 times the first peak, as shown in Figure 9, does it mean that the deformation of the spring has also changed? Is it reasonable to assuming that F0 is constant, as described in line 132. 

8. Does the fluid fill the cavity to the initial state after a completed cycle? If so, how to explain the changes in pressure and deformation with time in Figure 9? 

9. The results of the external elastic force with volume and velocity should be analyzed quantitatively.

Comments on the Quality of English Language

~~~

Author Response

(The authors gave the same response as above.)

Round 2

Reviewer 1 Report

Comments and Suggestions for Authors

Dear Authors,

I have no further comments related to revised version of the manuscript.

It can be published after minor revision (methodological errors and text editing).

Kind Regards 

Comments on the Quality of English Language

Moderate editing of English language required